# Single-Cell Transcriptomics Reveals Early Effects of Ionizing Radiation on Bone Marrow Mononuclear Cells in Mice

**DOI:** 10.3390/ijms25179287

**Published:** 2024-08-27

**Authors:** Yun-Qiang Wu, Ke-Xin Ding, Zhi-Chun Lv, Zheng-Yue Cao, Ke Zhao, Hui-Ying Gao, Hui-Ying Sun, Jing-Jing Li, Si-Yu Li, Xiong-Wei Zhao, Yang Xue, Shen-Si Xiang, Xiao-Fei Zheng, Xiao-Ming Yang, Chang-Yan Li

**Affiliations:** 1State Key Laboratory of Proteomics, National Center for Protein Sciences (Beijing), Beijing Institute of Radiation Medicine, Beijing 100850, China; wuyunqiang0512@163.com (Y.-Q.W.); xfzheng100@126.com (X.-F.Z.); 2School of Basic Medicine, Anhui Medical University, Hefei 230032, China; 3School of Life Sciences, Hebei University, No. 180 Wusi Dong Road, Lian Chi District, Baoding 071000, China

**Keywords:** irradiation, single-cell transcriptome sequencing, bone marrow mononuclear cells, radiosensitivity

## Abstract

Ionizing radiation exposure can cause damage to diverse tissues and organs, with the hematopoietic system being the most sensitive. However, limited information is available regarding the radiosensitivity of various hematopoietic cell populations in the bone marrow due to the high heterogeneity of the hematopoietic system. In this study, we observed that granulocyte–macrophage progenitors, hematopoietic stem/progenitor cells, and B cells within the bone marrow showed the highest sensitivity, exhibiting a rapid decrease in cell numbers following irradiation. Nonetheless, neutrophils, natural killer (NK) cells, T cells, and dendritic cells demonstrated a certain degree of radioresistance, with neutrophils exhibiting the most pronounced resistance. By employing single-cell transcriptome sequencing, we investigated the early responsive genes in various cell types following irradiation, revealing that distinct gene expression profiles emerged between radiosensitive and radioresistant cells. In B cells, radiation exposure led to a specific upregulation of genes associated with mitochondrial respiratory chain complexes, suggesting a connection between these complexes and cell radiosensitivity. In neutrophils, radiation exposure resulted in fewer gene alterations, indicating their potential for distinct mechanisms in radiation resistance. Collectively, this study provides insights into the molecular mechanism for the heterogeneity of radiosensitivity among the various bone marrow hematopoietic cell populations.

## 1. Introduction

People are frequently exposed to various radiation environments, such as natural background radiation, radiological examinations, and even nuclear disasters like the Chernobyl and Fukushima nuclear power plant accidents. Ionizing radiation can cause damage to diverse tissues and organs, with the hematopoietic system being the most sensitive [1]. Studies have previously investigated the damage patterns of white blood cells in peripheral blood after exposure [2,3]. It was found that radiation doses exceeding 2 Gy can result in a significant reduction in the number of blood cells. After exposure to 5–10 Gy, the number of lymphocytes can decrease by 50% within 24 h, with a more severe decline within 48 h [4]. In fact, there are numerous cell types in the hematopoietic system that exhibit strong heterogeneity, and different cell types vary greatly in their sensitivity to radiation. It has been reported that B cells are the most sensitive to radiation, while T cells exhibit intermediate sensitivity in peripheral blood. Natural killer cells, neutrophils, monocytes, and platelets demonstrate some degree of radiation resistance [5]. However, limited information is available regarding the radiosensitivity of the cell populations in bone marrow. Furthermore, the molecular mechanism underlying the differences in cellular radiosensitivity remains unclear.

In this study, by employing single-cell RNA-Seq, we investigated the early responsive genes in various cell types in bone marrow following irradiation, revealing that distinct gene expression profiles emerged between radiosensitive and radioresistant cells. The results of this study will enhance our comprehension of the molecular mechanisms underlying the heterogeneity of radiosensitivity in hematopoietic cell populations.

## 2. Results

### 2.1. Radiosensitivity Analysis of Multiple Cellular Populations in Bone Marrow (BM)

To evaluate the radiosensitivity of diverse bone marrow cell types, the mice were exposed to a single whole-body dose of 6.0 Gy from a ^60^Co γ source, and the cell populations’ frequencies and counts in BM were monitored at different time points. It was observed that at 6 h post-IR, the number of B cells decreased by about 70%. At 12 h post-IR, only about 10% of B cells were detected (Figure 1A,B). The number of hematopoietic stem progenitor cells (HSPCs) and granulocyte–macrophage progenitors (GMPs) also declined dramatically at 6 h. We also found a noticeable decline in the monocyte count at 12 h post-IR. However, the numbers of neutrophils, NKs, T cells, and dendritic cells (DCs) did not exhibit significant changes at 12 h post-IR (Figure 1C,D). To confirm this result, we investigated the numbers of the indicated cell populations in BM at a lower whole-body dose (4 Gy) IR. Consistently, similar results were obtained (Figure 1E,H). These results indicate that in the bone marrow, GMP, B cells, and HSPCs exhibit higher sensitivity to radiation, monocytes show moderate radiosensitivity, while neutrophils, NK cells, T cells, and dendritic cells display varying degrees of radiation resistance.

Previous studies suggested that mature neutrophils are present in the vasculature in two pools: a free-flowing intravascular blood pool (peripheral blood, PB) and a marginated pool residing in certain tissues, such as the liver, spleen, and bone marrow [6]. To confirm the irradiation resistance of neutrophils, we investigated the neutrophil numbers in the PB, spleen, and liver (Figure 1I). Consistently, the neutrophils in these pools significantly increased, especially at 6 h post-IR, indicating that neutrophils from different pools displayed consistent irradiation resistance.

### 2.2. Single-Cell Transcriptomes of Murine BM Mononuclear Cells after IR

To gain a comprehensive understanding of the heterogeneous impact of irradiation on the hematopoietic cellular composition in mouse bone marrow, scRNA-Seq was conducted in the NC group and the IR group (Figure 2A). After rigorous quality control, filtering, and the removal of cell doublets, we obtained 4709 cells and 3118 genes in the NC group and 4729 cells and 3048 genes in the IR group. Based on the differential expression of marker genes, we identified and visualized 20 clusters using a uniform manifold approximation and projection (UMAP) (Figure 2B). Each cluster was then annotated using the CellMarker database, the Cell Taxonomy database, and marker genes reported in the previous literature [7,8,9] (Figure 2C). We identified a total of 11 cell populations, including HSPCs, GMP, monocyte/monocyte-derived macrophage (Mo/MoDM), neutrophil, Basophil, Eosinophil, T cells, B cells, NKs, DCs, and Erythrocytes (Ery) (Figure 2D).

We further analyzed the effects of IR on cell populations in BM. The split UMAPs showed that in IR murine BM, B cells were dramatically decreased (Figure 2E). The percentage of B cells decreased significantly from 21.62% before irradiation to a mere 2.05% after irradiation. In contrast, the percentage of neutrophils increased significantly from 36.32% before irradiation to 48.36% after irradiation, with T cells increasing from 8.43% to 12.56% and NK cells increasing from 4.90% to 8.69%. (Figure 2F, Appendix A).

The analysis of the deferentially expressed genes (DEGs) in these cell populations suggested that radiation exposure was able to induce significant transcriptional differences across all cell populations (Figure 2G). B cells demonstrated the most drastic transcriptome alterations, whereas neutrophils, NK cells, and GMPs exhibited fewer DEGs with radiation exposure (Figure 2H), indicating that there was significant heterogeneity in the response to ionizing radiation among different cell subtypes in the bone marrow, both at the cellular and transcriptional levels.

### 2.3. Early Response Genes in Radiosensitive Cells 

The above results show that GMP, HSPCs, and B cells exhibit obvious sensitivity to IR. We first analyzed the early response genes in HSPCs following IR. HSPCs were defined (Figure 3A) and 88 DEGs, including 66 upregulated genes and 22 downregulated genes, were identified (Appendix A, Figure 3B). It was not feasible for downregulated genes to enrich the relevant biological processes due to the limited numbers. The upregulated genes were significantly enriched in the apoptotic process, DNA damage response, inflammatory response, negative regulation of cell growth, and positive regulation of reactive oxygen species metabolic process (Figure 3C). In GMPs (Figure 3D), thirty upregulated genes and nine downregulated genes were identified (Appendix A, Figure 3E). The GO analysis of the upregulated genes primarily revealed enrichment in the positive regulation of the intrinsic apoptotic signaling pathway, inflammatory response, and positive regulation of the reactive oxygen species (ROS) metabolic process (Figure 3F).

It is interesting to note that, compared to other mature cells, B cells in the bone marrow exhibit the most pronounced sensitivity to IR. In B cells (Figure 3G), 116 upregulated genes and 179 downregulated genes (Appendix A, Figure 3H) were identified. GO and KEGG analysis indicated that the upregulated genes were significantly enriched in the negative regulation of cell growth, apoptotic process, response to DNA damage, and inflammatory response, and that the downregulated genes were primarily associated with the B cell receptor signaling pathway, AMPK signaling pathway, and NF-κB signaling pathway (Figure 3I).

In Mo/MoDM (Figure 3J), 72 upregulated genes and 19 downregulated genes were identified (Appendix A, Figure 3K). The upregulated genes were primarily enriched in the inflammatory response, apoptotic process, and positive regulation of reactive oxygen species metabolic process (Figure 3L).

Based on the above results, the upregulated genes in these radiosensitive cells were primarily enriched in pathways related to apoptosis, inflammatory response, and ROS regulation. In addition, the common upregulated genes in these radiosensitive cells were further analyzed, and a total of 11 genes were obtained, including Exoc4, Mgmt, Gm42047, Ccng1, Chil3, Cox6b2, Bax, Rps27l, Dcxr, S100a8, and Lcn2 (Figure 3M). Among these, Rps27l, Bax, S100a8, Lcn2, and Mgmt are involved in the regulation of cell apoptosis. Notably, in B cells, several genes associated with mitochondrial respiratory chain complexes were specifically increased, including mt-Atp6, mt-Nd4, mt-Co2, mt-Cytb, mt-Co3, mt-Nd2, Cox6b2, mt-Nd3, and mt-Nd1 (Figure 3N).

### 2.4. Early Response Genes in Radioresistant Cells

Flow cytometry analysis suggested that neutrophils, NKs, T cells, and DCs exhibited a lower sensitivity to IR. The analysis of early response genes within neutrophils revealed a limited number of DEGs, with 24 upregulated genes and 39 downregulated genes (Appendix A, Figure 4B). The upregulated genes did not exhibit significant GO enrichment, while the downregulated genes were primarily enriched in the inflammatory response, chemotaxis, and positive regulation of the apoptotic process (Figure 4C).

The analysis of the DEGs in T cells (Figure 4D) revealed 71 upregulated genes and 21 downregulated genes (Appendix A, Figure 4E). GO analysis indicated that the upregulated genes were primarily enriched in the apoptotic process, cellular response to DNA damage stimulus, and negative regulation of cell growth (Figure 4F).

The analysis of the DEGs in NKs (Figure 4G) revealed 40 upregulated genes and 18 downregulated genes (Appendix A, Figure 4H). The upregulated genes were primarily enriched in the negative regulation of the apoptotic process, negative regulation of cell proliferation, and cellular response to DNA damage stimulus (Figure 4I).

Interestingly, compared to other radioresistant cell types, DCs exhibited much more DEGs at the early stage after radiation exposure, although the cell number of DCs was not obviously affected (Figure 4J). Specifically, 108 upregulated genes and 102 downregulated genes were identified (Appendix A, Figure 4K). GO analysis revealed that the upregulated genes were primarily enriched in the positive regulation of the apoptotic process, response to wounding, negative regulation of cell growth, Notch signaling pathway, and cellular response to DNA damage stimulus. The downregulated genes were primarily enriched in the adaptive immune response, including the positive regulation of natural killer cell-mediated cytotoxicity, positive regulation of CD8-positive, alpha-beta T cell activation, positive regulation of antibody-dependent cellular cytotoxicity, and antigen processing and presentation. Additionally, genes related to the positive regulation of NF-κB signaling were also downregulated (Figure 4L).

Due to the limited number of upregulated genes in neutrophils, we analyzed the common upregulated genes in NKs, T cells, and DCs. A total of 19 were obtained (Figure 4M), including Exoc4, Phlda3, Cdkn1a, Ccng1, Mgmt, Bax, Pvt1, Aen, A330023F24Rik, Ei24, Bbc3, Ptp4a3, Glipr1, Rps27l, Gm42047, Serpine2, Dglucy, Csnk1g1, and Trp53inp1. Among these genes, Cdkn1a, Mgmt, Bax, Aen, and Bbc3 are involved in regulating DNA damage response, while Ei24, Bax, Aen, Phlda3, Bbc3, and Trp53inp1 are implicated in the regulation of apoptosis. Furthermore, we noticed that in neutrophils, the majority of the DEGs were downregulated, and there were relatively fewer common DEGs compared to the other three cell populations. This suggested that neutrophils might possess unique mechanisms in resisting irradiation.

## 3. Discussion

Despite the widespread use of ionizing radiation in therapy and diagnostics and the inevitable exposure to external radiation, our understanding of radiation sensitivity in human blood cell populations remains limited, and the published data on this subject are inconsistent and varied. No comprehensive study has been conducted to systematically investigate the sensitivity of various bone marrow cell populations to IR. Additionally, the molecular mechanism for cell heterogeneity in radiation sensitivity remains unclear. In this study, we found that GMPs, HSPCs, and B cells in the bone marrow were the most radiosensitive cells, and neutrophils, NKs, T cells, and DCs demonstrated a certain degree of radiation resistance. Single-cell RNA-Seq analysis revealed that distinct gene expression profiles emerged between radiosensitive and radioresistant cells, indicating distinct responses to radiation exposure. This study provides insights into the molecular mechanism for the heterogeneity of radiosensitivity among the bone marrow cells.

Previous studies have confirmed that bone marrow hematopoietic progenitors are extremely sensitive to IR [10]. Due to their presence in the cell cycle, HSPCs are susceptible to radiation damage. However, it is intriguing that B cells, despite being mature blood cells generally not in the cell cycle, exhibit significant radiosensitivity. This study reveals that the number of B cells in the bone marrow decreases significantly at 6 h following irradiation, and 295 DEGs in B cells were identified. Among these DEGs, various radioresistant genes were downregulated, such as genes related to the NF-κB signaling pathway, including Nfkbia, Lyn, Traf3, Traf6, Blnk, Tnfrsf13c, Bcl2l1, Relb, and Birc3. Studies have demonstrated that the constitutive activation of NF-κB-associated genes in tumor cells can enhance their resistance to radiation [11]. For example, the BCL2L1 gene plays a crucial role in cell survival following radiation exposure, and the inhibition of BCL2L1 combined with radiotherapy significantly hinders tumor growth in vitro and in vivo [12]. In addition, genes associated with the AMPK signaling pathway are also significantly downregulated, including Pfkfb3, Tbc1d1, Eef2k, Pik3ca, Rps6kb1, Ppp2r2d, Akt3, Ppp2r5a, Pik3r1, Foxo3, Foxo1, etc. Research indicates that pharmacological PFKFB3 inhibition induces radiosensitization in transformed cells [13]. The overexpression of eEF2K led to radioresistance, and silencing eEF2K promoted radiosensitivity and apoptosis [14]. More interestingly, among the upregulated genes, genes related to mitochondrial respiratory chain complexes were also enriched, such as mt-Atp6, mt-Nd4, mt-Co2, mt-Cytb, mt-Co3, mt-Nd2, Cox6b2, mt-Nd3, and mt-Nd1. These proteins are components of mitochondrial complex I, III, IV, and V. Research has shown that mt-ATP6, mt-Nd1, mt-Nd5, and mt-Nd6 are upregulated in cells directly exposed to IR, suggesting that the mitochondrial gene expression response is part of a complex stress response operating in radiation-treated cells [15]. The upregulation of mitochondrial complex-related genes suggests that there may be higher levels of oxidative phosphorylation in B cells after irradiation. Although there is no evidence yet to demonstrate the association between mitochondrial complex activity and radiosensitivity, some studies have shown that anti-tumor compounds like chrysin can induce apoptosis in chronic lymphocytic leukemia B-lymphocytes by targeting mitochondrial complexes II and V, and the abnormal activation of mitochondrial complexes may render B cells more susceptible to exogenous toxins [16]. To illustrate the association between mitochondrial complexes and radiation sensitivity, further detailed investigations are necessary.

Our research demonstrates that neutrophils in the bone marrow exhibit significant radioresistance. Neutrophils, the most abundant cell type among white blood cells, possess a short lifespan and strong regenerative capacity [17]. Since neutrophils originate from HSPCs and GMPs located within the bone marrow and our data suggest that the numbers of HSPCs and GMPs declined significantly after irradiation, we believe that the lack of significant changes in the number of neutrophils may not be due to an increase in their production. Furthermore, a previous study suggested that the residence time of neutrophils in bone marrow is 0.7 days and the average half-life of circulating neutrophils is 12.5 h [18]. In our study, we observed a stable number of neutrophils within 12 h after irradiation, indicating that neutrophil clearance and destruction may not be a major factor in this process. Moreover, we found that the number of neutrophils in peripheral blood and margination pools increased significantly. Early studies suggested that neutrophils from either pool were indistinguishable [19,20]. These data suggested that the neutrophils in different pools displayed similar irradiation resistance. Studies have reported that peripheral blood neutrophils lack DNA damage repair responses following IR, such as the upregulation of γH2AX and the co-localization of MDC1, NBS1, MRE11, RAD50, ATM kinase, and 53BP1 with γH2AX [21]. DNA damage repair proteins, such as DNA-PKCs, ATR, and MGMT, are barely detectable in neutrophils. Correspondingly, neutrophils do not undergo radiation-induced cell death within 24 h post-IR. Our results showed that neutrophils only exhibited a few DEGs in the early stage post-IR. The genes associated with apoptosis, such as Gadd45b, Inpp5d, Ddit3, Ctsd, and Gadd45g, were downregulated. Some research has suggested that the high degree of chromatin condensation in neutrophils might play a significant role in their resistance to DNA damage induced by radiation. This could explain why these cells do not exhibit significant changes in DEGs following IR. Elucidating the molecular mechanisms underlying the radioresistance of neutrophils will help develop new targets for radioprotective drugs, thereby enhancing the body’s ability to resist irradiation.

In our study, a high degree of responsiveness of DCs to IR was observed at the transcriptome level. DCs are the most potent professional antigen-presenting cells and inducers of T cell-mediated immunity. The analysis of the DEGs revealed that upregulated genes were enriched in the Notch signaling pathway, including Notch1, Ptp4a3, Rps19, Bloc1s2, and Maml3. Studies have reported that Notch1 expression can be induced by high-dose IR [22], while inhibiting Notch1 can enhance the radiosensitivity of tumor cells [23]. In addition, genes related to response to wounding were also upregulated, such as Serpine2, Snhg15, Zfp36, Pvt1, Bax, and Sulf2. Previous studies reported that SERPINE2 can regulate the DNA damage response induced by IR in lung cancer and that knocking down SERPINE2 leads to abnormal DNA damage repair, resulting in radiation-induced cell death [24]. Furthermore, it was observed that several genes related to adaptive immune response were downregulated. It is crucial to elucidate the role of these genes in enabling DCs to resist radiation damage.

## 4. Materials and Methods

### 4.1. Mice

Male C57BL/6J mice were obtained from SPF (Beijing, China) and used at 8 to 10 weeks of age. Mice were housed under specific pathogen-free conditions. Mice were randomly assigned into a normal control (NC) group and an irradiated (IR) group. The IR group received a single whole-body dose of 6 Gy or 4 Gy from a ^60^Co γ source at a dose rate of 64.91 R/min, while the NC group did not receive any irradiation. At different times post-irradiation, mice were euthanized under anesthesia with sodium pentobarbital for experimental procedures.

### 4.2. Isolation of Bone Marrow Mononuclear Cells

The bone marrow cells were flushed from intact femurs and tibias and then mashed with a 2 mL disposable syringe plunger to generate a single-cell suspension. The collection of the cells was performed in 1640 media with 2% FBS and filtered through a 40 μm strainer. After centrifugation at 800× *g* for 5 min at 4 °C, the cell pellet was resuspended in 2 mL of RBC lysis buffer for 5 min. Centrifugation was conducted again to obtain the bone marrow mononuclear cells.

### 4.3. Flow Cytometry

Anti-CD45-PE, anti-CD3-APC, anti-CD11b-BV605, anti-B220-Pecy7, anti-NK1.1-APCH7, anti-CD11c-ef450, anti-F4/80-FITC, Ly6g-APC, Ly6c-Pecy7, APCH7-streptavidin, anti-Sca1-BV605, anti-cKit-Pecy7, anti-CD150-BV421, anti-CD48-AF700, anti-CD34-BV421, anti-CD16/32-AF700, RBC Lysis Buffer (420301), and biotin-conjugated mouse lineage cocktails were purchased from BioLegend (San Diego, CA, USA). Bone marrow mononuclear cells were stained with the indicated antibodies for 15 min. FACS analysis was performed on LSR-Fortessa, and data were analyzed using FlowJo 10.8.1 software.

### 4.4. Single-Cell Transcriptomic Analysis

The bone marrow single-cell suspension was adjusted to a concentration of 1 × 10^6 cells/mL; then, the quality control was performed before scRNA-Seq. Library preparation involved the use of the 10× Genomics Chromium Controller machine and Chromium Next GEM Single Cell 3′ Reagent Kits v3.1 (Dual Index). Each sample was processed for oil–water emulsion generation with 16,000 cells, followed by RT-PCR amplification, cDNA amplification, and library construction. Sequencing was carried out using the Qubit 4.0 machine with Qubit™ 1× dsDNA Assay Kits, a high sensitivity kit for library quality and concentration assessment, the StepOnePlus™ Real-Time PCR System for library molar concentration determination, LabChip Touch for library insert size detection, and Illumina’s NovaSeq 6000 sequencing platform for PE150 sequencing. To remove low-quality cells and likely multiplet captures, a major concern in microdroplet-based experiments, we applied criteria to filter out cells with gene numbers less than 200 and more than 10,000 and with unique molecular identifiers (UMIs) less than 1000. We further discarded low-quality cells where >30% of the counts belonged to mitochondrial genes. After applying these QC criteria, 4709 single cells from the NC group and 4729 from the IR group were included in downstream analyses. Data analysis included the use of Cell Ranger 6.0.1 to generate matrix files from the raw data, Seurat for cell expression quantification and clustering analysis, SingleR for cell annotation, and clusterProfiler for GO enrichment analysis and KEGG analysis.

### 4.5. Statistical Analysis

The experimental data were presented as mean ± standard deviation, with statistical significance indicated by *p* < 0.05. Statistical analysis was performed using the GraphPad Prism 9.0 software. FlowJo 10.8.1 software was used to analyze and graph the data values.

## 5. Conclusions

In summary, our study revealed the radiosensitivity heterogeneity of bone marrow cell populations. The results of this study contribute to a better understanding of the biological response mechanisms of bone marrow cells to IR and might provide potential candidate intervention targets for protection against radiation damage.

## Figures and Tables

**Figure 1 ijms-25-09287-f001:**
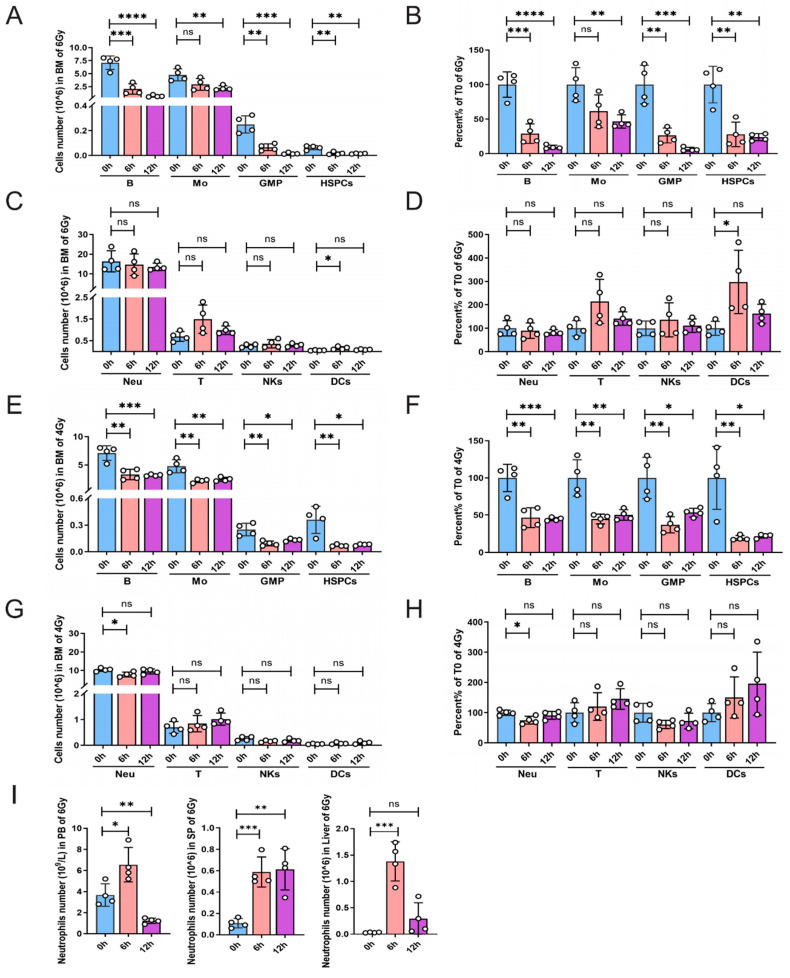
Radiosensitivity analysis of multiple cellular populations in BM. Mice were exposed to a single whole-body dose of 6 Gy (**A**–**D**) or 4 Gy (**E**–**H**) irradiation and, at the indicated time points, the BM mononuclear cells were analyzed by flow cytometry (*n* = 4). (**I**) Mice were exposed to a single whole-body dose of 6 Gy and, at the indicated time points, the neutrophil numbers were measured in the PB, spleen (SP), and liver. The following markers were used: HSPCs: Lin^−^Sca-1^+^c-Kit^+^; GMP: Lin^−^Sca-1^+^c-Kit^+^CD150^+^CD48^+^; B cells: B220^+^; T cells: CD3^+^; neutrophils (Neu): CD11b^+^Ly6g^+^; monocytes (Mo): CD11b^+^Ly6c^+^; NKs: NK1.1^+^; and DCs: CD11c^+^. Error bars represent SEM. * *p* < 0.05, ** *p* < 0.01, *** *p* < 0.001, **** *p* < 0.0001, ns: not significant.

**Figure 2 ijms-25-09287-f002:**
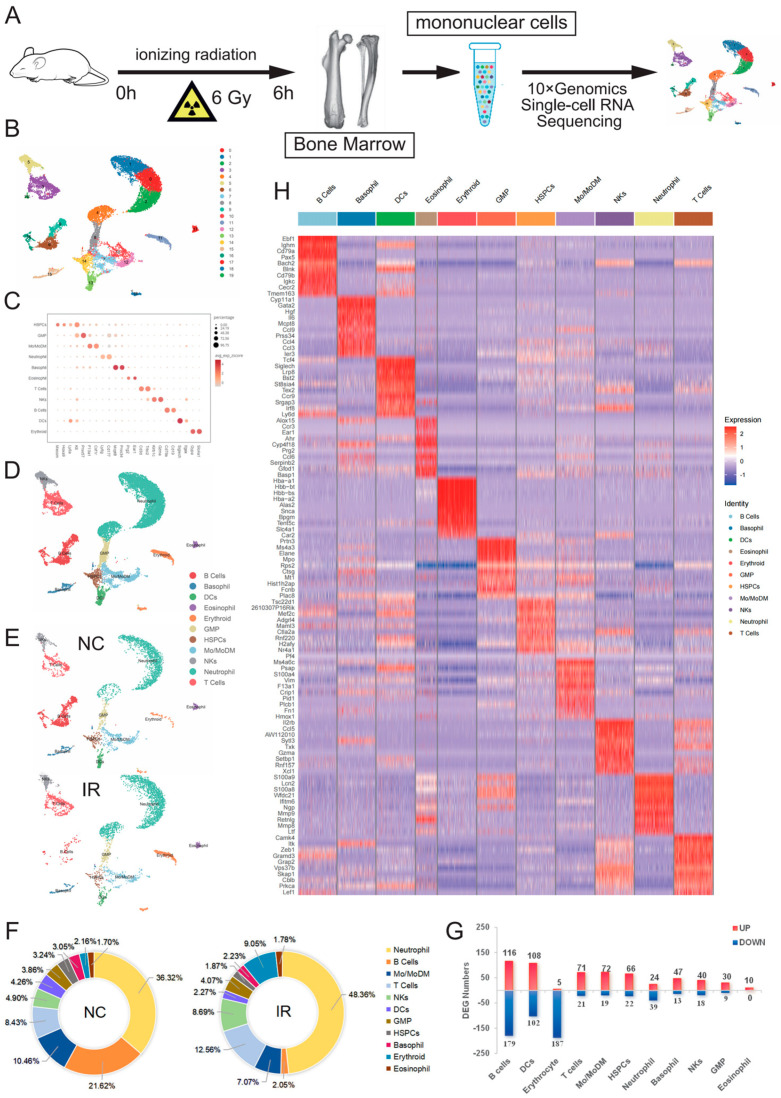
Single-cell transcriptomes of murine BM mononuclear cells after irradiation. (**A**) Schematic diagram of the experimental design. (**B**) UMAP plots (colored by clusters) of BM mononuclear cells from normal control (NC) and irradiated mice (IR). (**C**) Dot plot of 11 cell types with unique signature gene expression profiles. (**D**) UMAP plots (colored by 11 cell types) of BM mononuclear cells. (**E**) Split UMAPs from the NC and IR groups. (**F**) Proportion of identified cell types between groups. (**G**) Heatmap of the top 10 DEGs in individual cell types. (**H**) Numbers of significant DEGs (fold change > 1.5, adj.p < 0.05) of 11 populations between the NC and IR groups.

**Figure 3 ijms-25-09287-f003:**
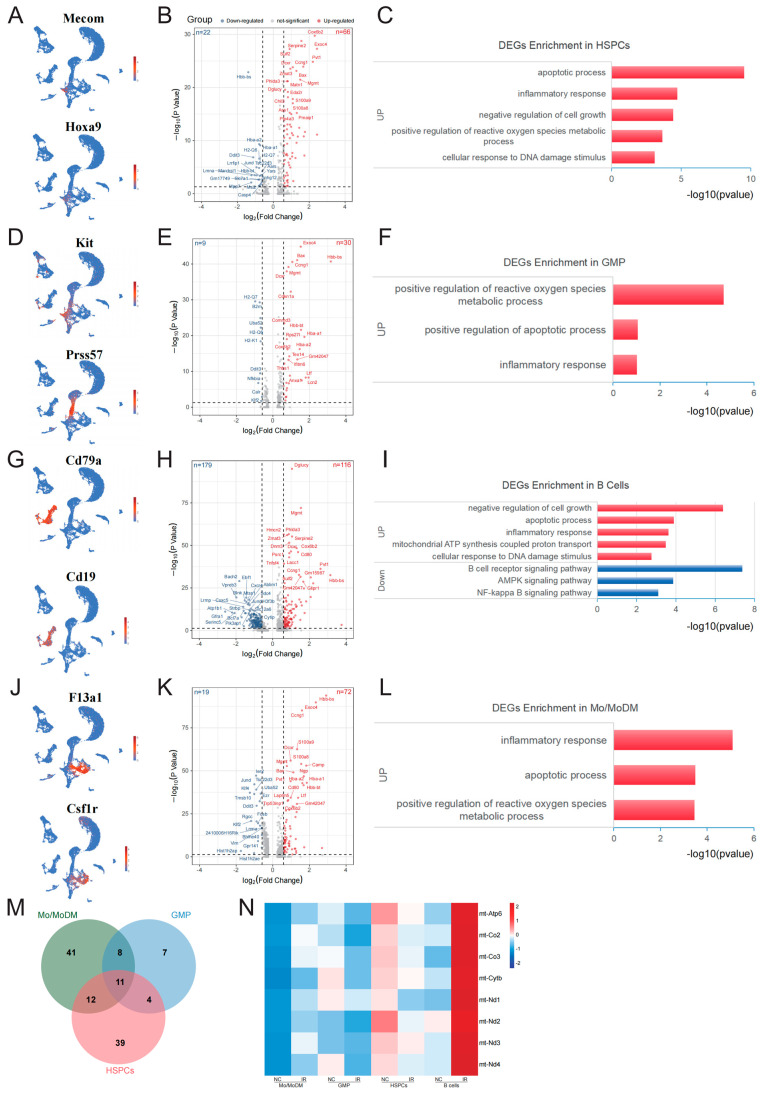
Early response genes in radiosensitive cells. (**A**) UMAP plot of specific markers defining HSPCs. (**B**) Volcano plot showing significant DEGs of HSPCs between the NC and IR groups. (**C**) The enrichment analysis of DEGs for HSPCs between the NC and IR groups. (**D**) UMAP plot of specific markers defining GMP. (**E**) Volcano plot showing significant DEGs of GMPs between the NC and IR groups. (**F**) The enrichment analysis of DEGs for GMPs between the NC and IR groups. (**G**) UMAP plot of specific markers defining B cells. (**H**) Volcano plot showing significant DEGs of B cells between the NC and IR groups. (**I**) The enrichment analysis of DEGs for B cells between the NC and IR groups. (**J**) UMAP plot of specific markers defining Mo/MoDM cells. (**K**) Volcano plot showing significant DEGs of Mo/MoDM cells between the NC and IR groups. (**L**) The enrichment analysis of DEGs for Mo/MoDM cells between the NC and IR groups. (**M**) Venn diagram showing the key genes in HSPCs, GMP, B cells, and Mo/MoDM. (**N**) Heatmap showing the expression of mitochondrial respiratory chain complex genes in HSPCs, GMP, B cells, and Mo/MoDM.

**Figure 4 ijms-25-09287-f004:**
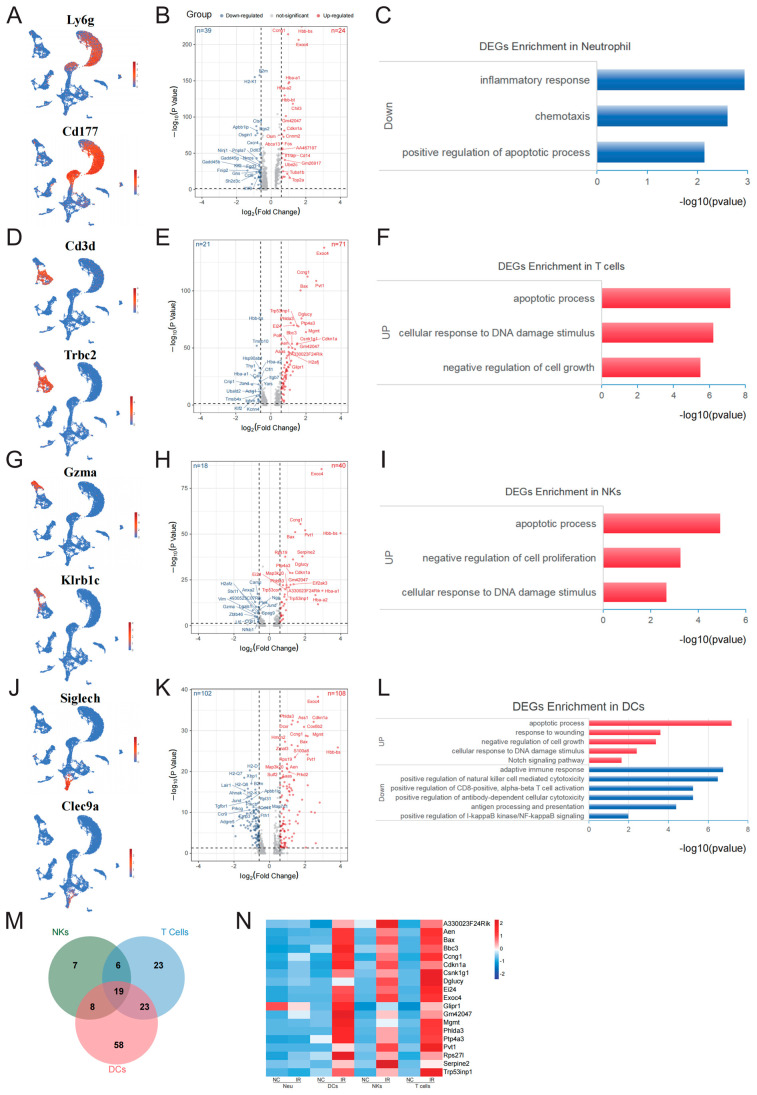
Early response genes in radioresistant cells. (**A**) UMAP plot of specific markers defining neutrophils. (**B**) Volcano plot showing significant DEGs of neutrophils between the NC and IR groups. (**C**) The enrichment analysis of DEGs for neutrophils between the NC and IR groups. (**D**) UMAP plot of specific markers defining T cells. (**E**) Volcano plot showing significant DEGs of T cells between the NC and IR groups. (**F**) The enrichment analysis of DEGs for T cells between the NC and IR groups. (**G**) UMAP plot of specific markers defining NKs. (**H**) Volcano plot showing significant DEGs of NKs between the NC and IR groups. (**I**) The enrichment analysis of DEGs for NKs between the NC and IR groups. (**J**) UMAP plot of specific markers defining DC. (**K**) Volcano plot showing significant DEGs of DC cells between the NC and IR groups. (**L**) The enrichment analysis of DEGs for DC cells between the NC and IR groups. (**M**) Venn diagram showing the key genes in T cells, NKs, and DC cells. (**N**) Heatmap showing the expression of key genes in T cells, NKs, and DC cells.

## Data Availability

Data contained within the article.

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
