# Peer review of "Single-Cell Transcriptomics Reveals Early Effects of Ionizing Radiation on Bone Marrow Mononuclear Cells in Mice"

_ijms, 2024, doi:10.3390/ijms25179287_

Round 1

Reviewer 1 Report

Comments and Suggestions for Authors

Exposure to ionizing radiation can damage tissues and organs, especially the hematopoietic system. In this study authors stated that granulocyte-macrophage progenitors, hematopoietic stem/progenitor cells, and B cells in the bone marrow were most sensitive to radiation. Moreover, scRNA-Seq revealed different gene expression profiles between radiosensitive and radioresistant cells. The study exploits a novel approach however there are some concerns regarding the methodology. 

  1. Authors do not specify the time and period or exposure. In the first figure, does the timing refer to the duration of the exposure or the time of recovery from a shorter time of exposure? 
  2. Why did the authors used 6 Gy of ionizing radiation? Are available results at lower intensity? This intensity is too high to be considered alone
  3. At Pag3, line 115-117:After rigorous quality control, filter and removal of cell doublets, we obtained 9438 cells and 3188 genes. Based on the differential expression of marker genes…” If the mentioned number of cells and genes are relative to a specific condition it has to be specified and both the control and treated condition should be mentioned. 
  4. The radiation resistant population (neutrophils, T cells, and NK) could be linked to the Neutrophil abundance, their brief (6–8 h) circulating half life, and intravascular margination, clearance and destruction. Did the author checked for this occurrence? It is incorrect calling them “resistant" without this demonstration. Authors should add experiments that are able to follow the treated population and to distinguish new cells from the treated ones. 
  5. The limited number of DEG neutrophils should be the consequence of point 4. In my opinion the analysis performed adding common upregulated genes in NKs, T cells, and DCs is not correct. (Page 7)
  6. Discussion and conclusions sections should be extensively revised after the solicited experimental part. 
Comments on the Quality of English Language

Text should be revised for typos.

Author Response

Comments 1: [Authors do not specify the time and period or exposure. In the first figure, does the timing refer to the duration of the exposure or the time of recovery from a shorter time of exposure?]

Response 1: In the revised manuscript, we gave a detailed description about the irradiation exposure condition in Materials and Methods part and figure legend of Figure 1.

At Page 2, Line 57-62: Mice were randomly assigned into a normal control group (NC) and irradiated groups (IR). IR group were exposed to a single whole-body dose of 6 Gy or 4 Gy from a 60Co γ source, at a dose rate of 64.91 R/min. Mice in the control group did not receive any irradiation. At different time post-irradiation, mice were euthanized under anesthesia with sodium pentobarbital for experimental procedures.

At Page 4, Line 127-129: Mice were exposed to a single whole-body dose of 6 Gy or 4 Gy irradiation and at the indicated time the BM mononuclear cells were analyzed by flow cytometry (n=4).

Comments 2: [Why did the authors used 6 Gy of ionizing radiation? Are available results at lower intensity? This intensity is too high to be considered alone.]

Response 2: Thanks a lot for the good suggestion. In the revised manuscript, we performed the investigations at 4 Gy irradiation and the similar results were obtained.

At Page 3, Line 113-116:To confirm this result, we investigated the numbers of the indicated cell population in BM at a lower whole-body dose (4 Gy) IR. Consistently, the similar results were obtained (Figure 1E-H).

Comments 3: [At Pag3, line 115-117: “After rigorous quality control, filter and removal of cell doublets, we obtained 9438 cells and 3188 genes. Based on the differential expression of marker genes…” If the mentioned number of cells and genes are relative to a specific condition it has to be specified and both the control and treated condition should be mentioned.]

Response 3: Thanks for the good suggestion. In the revised manuscript, we added the information just as the reviewer suggested.

At Page 2, Line 90-96: To remove low-quality cells and likely multiplet captures, a major concern in micro-droplet-based experiments, we applied criteria to filter out cells with gene numbers less than 200 and more than 10000, unique molecular identifier (UMI) less than 1000. We further discarded low-quality cells where >30% of the counts belonged to mitochondrial genes. After applying these QC criteria, 4709 single cells from the NC group and 4729 from the IR group were included in downstream analyses.

Comments 4: [The radiation resistant population (neutrophils, T cells, and NK) could be linked to the Neutrophil abundance, their brief (6–8 h) circulating half life, and intravascular margination, clearance and destruction. Did the author checked for this occurrence? It is incorrect calling them “resistant" without this demonstration. Authors should add experiments that are able to follow the treated population and to distinguish new cells from the treated ones.]

Response 4: Thanks a lot for the very good suggestions! We carefully analyzed our own results and performed further measurement based on three considerations. First, because the neutrophil originates from HSPCs and GMPs located within the bone marrow and considering that the numbers of HSPCs and GMPs declined significantly after irradiation, we believed that lack of significant change in the number of neutrophils may not be due to an increase in their production. Second, a previous study suggested that the residence time of neutrophils in bone marrow is 0.7 days and the average half-life of circulating neutrophils was 12.5 hours. In our study, we observed a stable number of neutrophils within 12 hours after irradiation, indicating that the clearance and destruction of neutrophils might not involve in this process. Third, just as the reviewer mentioned, mature neutrophils are present in the vasculature in two pools: a free-flowing intravascular blood pool (peripheral blood) and a blood pool residing in certain tissues (marginated pool). The major sites for marginated neutrophils in humans are the liver, spleen and bone marrow itself. So, we investigated the neutrophils number in PB, spleen and liver in the revised manuscript. Consistently, the neutrophils in these pools displayed significant resistance as the BM neutrophils did. All these data indicated that neutrophil is a resistant cell population. Also in the discussion part, we gave some discussion about this result.

At Page 3, Line 119-125: Previous studied suggested that mature neutrophils are present in the vasculature in two pools: a free-flowing intravascular blood pool (peripheral blood, PB) and a marginated pool residing in certain tissues such as the liver, spleen and bone marrow [6]. To confirm the irradiation resistance of neutrophils, we investigated the neutrophils number in PB, spleen and liver (Figure 1I). Consistently, the neutrophils in these pools significantly increased especially at 6 hours post-IR, indicating that neutrophils from different pools displayed consistent irradiation resistance.

Comments 5: [The limited number of DEG neutrophils should be the consequence of point 4. In my opinion the analysis performed adding common upregulated genes in NKs, T cells, and DCs is not correct. (Page 7)]

Response 5: Thanks for the suggestion. Just as described in response to point 4, we hypothesize that the limited number of DEG in neutrophils might be due to the resistance. However, through analyzing the DEGs in different cells populations, we believed that the mechanism of neutrophil in resisting radiation may not be entirely the same as that of other cells since the number of DEGs in neutrophil is limited. So, in order to reveal the common mechanisms of radiation resistance in NKs, T cells and DCs, we analyzed the common upregulated genes in these cell populations.

Comments 6: [Discussion and conclusions sections should be extensively revised after the solicited experimental part.]

Response 6: In the revised manuscript, we made modifications on the discussion part just as the reviewer suggested.

At Page 11, Line 309-320: Since neutrophil originates from HSPCs and GMPs located within the bone marrow and our data suggest that the numbers of HSPCs and GMPs declined significantly after irradiation, we believed that lack of significant change in the number of neutrophils may not be due to an increase in their production. Furthermore, a previous study suggested that the residence time of neutrophils in bone marrow is 0.7 days, and the average half-life of circulating neutrophils was 12.5 hours [17]. In our study, we observed a stable number of neutrophils within 12 hours after irradiation, indicating that the clearance and destruction of neutrophils might not involve in this process. Moreover, we found that the number of neutrophils in peripheral blood and margination pools increased significantly. Early studies suggested that neutrophils from either pool were indistinguishable [18, 19]. These data suggest that the neutrophils in different pools displayed similar irradiation resistance.

At Page 12, Line 331-333: Elucidating the molecular mechanisms underlying the radioresistance of neutrophil will help develop new targets for radioprotective drugs, thereby enhancing the body's ability to resist irradiation.

Reviewer 2 Report

Comments and Suggestions for Authors

The manuscript presents an elegant study of the effects early radiation effects on different bone marrow mononuclear cells in mice after whole-body irradiation. This study, presented different levels of radiosensitivity for each cellular type and single-cell transcriptomic sequencing was applied to mapping differential gene expression profiles in each population that ultimately can correlate with each cell-type intrinsic radiationsensitity.
The  manuscript is well written, however, I suggest the authors add some more details about the experimental setup as follows:

1-  Materials and Methods: Mice: It is not clear from the text the number of animals used for each condition (irradiated and control)

2 - Materials and Methods: Irradiation setup should be described, such as dose-rate, enenergy and so on. 

3 - Results 3.1: It should be stated in the text that 6Gy were administrated as a whole-body irradiation, in a similar manner of the FIgure 1 legend.

4 - Figure 1: From the legend and the figure itself it is not clear to me what the percentage in plot A and C refers to: is this total number of cells at T=0?

5 - Results:  3.2: Lines 127 and 128, Authors should state numbers here too, as done previously for B cells.

6- Discussion: Lines 252 and 253, needs a reference

7- Discussion: Lines 296-298 this sentence is not clear. Neutrophils are inserted in the group of radioresistant cells, with a stable cellular population in the first 12 hours (Figure 1). Understanding their molecular mechanisms of resistance would probably be better applied to increase/adapt the radiation response of more radiosensitive cells.

8 - Conclusions: Authors stated: "... provide important insights for radiation protection" This sentence is vague would be nice if authors could clarify what are/is the important insights obtained from this study.  

Author Response

Comments 1: [Materials and Methods: Mice: It is not clear from the text the number of animals used for each condition (irradiated and control).]

Response 1: Thanks a lot for the suggestion. In the revised manuscript, we described the number of mice used for each condition (irradiated and control).

At Page 4, Line 127-129: Mice were exposed to a single whole-body dose of 6 Gy (A-D) or 4 Gy (E-H) irradiation and at the indicated time points, the BM mononuclear cells were analyzed by flow cytometry (n=4).

Comments 2: [Materials and Methods: Irradiation setup should be described, such as dose-rate, energy and so on.]

Response 2: Thanks for the important notification. In the revised manuscript, we added the detailed information about the irradiation condition.

At Page 2, Line 57-62: Mice were randomly assigned into a normal control group (NC) and irradiated groups (IR). IR group were exposed to a single whole-body dose of 6.0 Gy or 4Gy from a 60Co γ source, at a dose rate of 64.91 R/min. Mice in the control group did not receive any irradiation. At different time post-irradiation, mice were euthanized under anesthesia with sodium pentobarbital for experimental procedures.

Comments 3: [Results 3.1: It should be stated in the text that 6Gy were administrated as a whole-body irradiation, in a similar manner of the Figure 1 legend.]

Response 3: Thanks for the suggestion. In the revised manuscript, we made modification just as the reviewer suggested.

At Page 3, Line 105-106: the mice were exposed to a single whole-body dose of 6 Gy from a 60Co γ source.

Comments 4: [Figure 1: From the legend and the figure itself it is not clear to me what the percentage in plot A and C refers to: is this total number of cells at T=0?]

Response 4: Thanks for the comment. In the revised manuscript, we analyzed the percentage of cell numbers in IR relative to NC (T0).

Comments 5: [Results: 3.2: Lines 127 and 128, Authors should state numbers here too, as done previously for B cells.]

Response 5: Thanks a lot for the good suggestion. In the revised manuscript, we added the information just as the reviewer suggested.

At Page 5, Line 149-151: the percentage of neutrophils increased significantly from 36.32% before irradiation to 48.36% after irradiation with T cells from 8.43% to 12.56%, and NK cells from 4.90% to 8.69%.

Comments 6: [Discussion: Lines 252 and 253, needs a reference.]

Response 6: In the revised manuscript, we added the reference.

Comments 7: [Discussion: Lines 296-298 this sentence is not clear. Neutrophils are inserted in the group of radioresistant cells, with a stable cellular population in the first 12 hours (Figure 1). Understanding their molecular mechanisms of resistance would probably be better applied to increase/adapt the radiation response of more radiosensitive cells.]

Response 7: Thanks for the suggestion. In the revised manuscript, we modified the description.

At Page 12, Line 331-333: Elucidating the molecular mechanisms underlying the radioresistance of neutrophil will help develop new targets for radioprotective drugs, thereby enhancing the body's ability to resist irradiation.

Comments 8: [Conclusions: Authors stated: "... provide important insights for radiation protection" This sentence is vague would be nice if authors could clarify what are/is the important insights obtained from this study.]

Response 8: In the revised manuscript, we modified this sentence.

At Page 12, Line 348-350:  The results of this study contribute to a better understanding of the biological response mechanisms of bone marrow cells to IR and might provide potential candidate intervention targets for protection against radiation damage.

Round 2

Reviewer 1 Report

Comments and Suggestions for Authors

Authors provided correct and exhaustive answers.